# Effects and Mechanisms of Tea Regulating Blood Pressure: Evidences and Promises

**DOI:** 10.3390/nu11051115

**Published:** 2019-05-18

**Authors:** Daxiang Li, Ruru Wang, Jinbao Huang, Qingshuang Cai, Chung S. Yang, Xiaochun Wan, Zhongwen Xie

**Affiliations:** 1State Key Laboratory of Tea Plant Biology and Utilization, School of Tea and Food Sciences and Technology, Anhui Agricultural University, Hefei 230036, China; dxli@ahau.edu.cn (D.L.); wrr@ahau.edu.cn (R.W.); jinbaohuang@ahau.deu.cn (J.H.); qingshuang@ahau.edu.cn (Q.C.); xcwan@ahau.edu.cn (X.W.); 2International Joint Laboratory on Tea Chemistry and Health Effects of Ministry of Education, Anhui Agricultural University, Hefei 230036, China; csyang@pharmacy.rutgers.edu; 3Department of Chemical Biology, Ernest Mario School of Pharmacy, Rutgers, The State University of New Jersey, Piscataway, NJ 08854-8020, USA

**Keywords:** tea secondary metabolites, hypertension, endothelial function, inflammation

## Abstract

Cardiovascular diseases have overtaken cancers as the number one cause of death. Hypertension is the most dangerous factor linked to deaths caused by cardiovascular diseases. Many researchers have reported that tea has anti-hypertensive effects in animals and humans. The aim of this review is to update the information on the anti-hypertensive effects of tea in human interventions and animal studies, and to summarize the underlying mechanisms, based on ex-vivo tissue and cell culture data. During recent years, an increasing number of human population studies have confirmed the beneficial effects of tea on hypertension. However, the optimal dose has not yet been established owing to differences in the extent of hypertension, and complicated social and genetic backgrounds of populations. Therefore, further large-scale investigations with longer terms of observation and tighter controls are needed to define optimal doses in subjects with varying degrees of hypertensive risk factors, and to determine differences in beneficial effects amongst diverse populations. Moreover, data from laboratory studies have shown that tea and its secondary metabolites have important roles in relaxing smooth muscle contraction, enhancing endothelial nitric oxide synthase activity, reducing vascular inflammation, inhibiting rennin activity, and anti-vascular oxidative stress. However, the exact molecular mechanisms of these activities remain to be elucidated.

## 1. Introduction

Cardiovascular diseases (CVDs) are a group of diseases of the heart and blood vessels that include coronary heart disease, cerebrovascular disease, peripheral arterial disease, rheumatic heart disease, congenital heart disease and deep vein thrombosis [1]. During recent years, CVDs have overtaken cancer as the leading cause of deaths worldwide [2]. However, most CVDs can be prevented by modifying risk factors such as imbalanced diet, physical inactivity, diabetes, elevated lipids and high blood pressure [3]. Of these risk factors, high blood pressure is the most dangerous factor linked to CVDs death events [4]. It is estimated that high blood pressure is a comorbid factor in 69% of people who have their first heart attack, and 75% of those with chronic heart failure [5]. Clinical data shows that a 5 mmHg blood pressure reduction can reduce the risk of stroke and ischemic heart disease by 34 and 21%, respectively [6,7].

High blood pressure, or hypertension, is diagnosed when an individual has a systolic blood pressure (SBP) of 130–139 mmHg or a diastolic blood pressure (DBP) of 80–89 mmHg (Stage 1) and a systolic blood pressure (SBP) of ≥140 mmHg and/or a diastolic blood pressure (DBP) of ≥90 mmHg (Stage 2). [8] A significant prevalence of high blood pressure among adults aged 25 years and older exists, with a worldwide incidence of 40% [5]. Aging, dietary factors (for example alcohol consumption, excessive salt intake, and insufficient fruit and vegetable consumption), lifestyle factors (such as smoking and physical inactivity), and genetic predisposition have all been implicated in developing hypertension [9]. It has been estimated that hypertension affects one billion people and causes 9.4 million deaths every year globally [3]. This toll continues to increase as the incidence of hypertension rises sharply in low- and middle-income countries, where the social and economic costs associated with the disease are expected to place an especially heavy burden on socioeconomic development [10].

Tea is a beverage prepared by pouring hot or boiling water over the cured leaves or leaf buds of the tea plant *Camellia sinensis*. Based on the degree of fermentation, tea can also be classified into three major types: unfermented green tea, fermented black tea, and semi-fermented Oolong tea [11]. Tea is the second most consumed beverage after water and is thought to have a variety of health benefits [11,12]. It contains characteristic polyphenolic compounds known as catechins, namely (-)-epicatechin (EC), (-)-epigallocatechin (EGC), (-)-epicatechin gallate (ECG), (-)-epigallocatechin gallate (EGCG), (+)-catechin (C), and (+)-gallocatechin (GC), plus a small amount of (-)-catechin gallate (CG) and (-)-gallocatechin gallate (GCG). [13] A number of studies have shown that consumption of both green and black teas is linked to reductions in the risk of CVDs and some forms of cancers, to improved oral health, weight gain control and cognition in the elderly, and to increased antibacterial and antiviral activity and bone density [14,15]. These health benefits are often attributed to tea being rich in a class of polyphenolic compounds called flavonoids [16]. Diet plays an important role in the treatment and control of high blood pressure [17,18]. A survey showed that flavonoids can play an important role in the treatment and control of high blood pressure [19]. Anti-hypertension effects of drinking tea have become a hot topic for molecular nutrition and food research. In order to better understand the research achievements in this field, we summarized the role of tea in lowering blood pressure in clinical studies, as well as animal and cell experiments. The molecular mechanisms of tea hypotension effects were updated in this review.

## 2. Tea Regulating Blood Pressure in Human Intervention Studies 

### 2.1. Hypotension Effects of Tea in Human Population Studies by Meta-analysis

In China, there is a long history of people drinking tea, made by collecting leaves from old tea plants to treat high blood pressure. Both in East Asian and western countries, lowering blood pressure by drinking tea has been reported in human population studies. Due to differences (genetic backgrounds, body composition, dietary habits, and amount and type of tea consumed) between different populations, the results of tea consumption in lowering blood pressure may not be consistent. However, during recent years, with improvements in experimental design and statistic software, great advances have been made in understanding the effect of tea consumption on blood pressure. Khalesi et al. [20] systematically reviewed randomized controlled trials that examined the effect of green tea consumption on blood pressure using meta-analysis. Based on their selected criteria, they collected papers from ProQuest, PubMed, Scopus and Cochrane Library published from 1995 to 2013. Thirteen studies were included for meta-analysis. The results showed that consumption of green tea significantly reduced SBP by 2.08 mmHg and DBP by 1.71 mmHg. In addition, subgroup analysis suggested a greater reduction in both SBP and DBP in participants whose baseline mean systolic blood pressure was ≥130 mmHg. Peng et al. [21] also investigated the effect of green tea consumption on blood pressure based on a meta-analysis of 13 randomized controlled trials across several countries, which were published in PubMed, Embase and Cochrane Library (up to March 2014). Thirteen trials containing 1,367 subjects were included for the meta-analysis. The results indicated that consumption of green tea significantly reduced SBP level by 1.98 mmHg. Compared with the control group, green tea also showed a significant lowering effect on DBP in the treatment group (1.92 mmHg). Subgroup analyses further suggested that the positive effect of green tea polyphenols on blood pressure occurred with a low dose of green tea polyphenols (<582.8 mg/day) with long-term duration (≥12 weeks), whilst ruling out the confounding effects of caffeine. This analysis also supports the premise that green tea consumption has a favorable effect of decreasing blood pressure. Black tea is more popular than green tea in western countries. Greyling et al. [22] investigated the effect of black tea consumption on blood pressure based on a meta-analysis of 11 randomized controlled trials. The literature was systematically collected from Medline, Biosis, Chemical Abstracts and EMBASE databases through July 2013. Eleven studies (12 intervention arms, 378 subjects, dose of 4–5 cups of tea) were included for analysis. The SBP and DBP were decreased 1.8 mmHg (*P* = 0.0013) and 1.3 mmHg (*P* < 0.0001), respectively, by regular tea ingestion. Liu et al. [23] evaluated the effect of both green and black tea intake on blood pressure by a meta-analysis of randomized controlled trials. A systematic search was conducted in MEDLINE, EMBASE and the Cochrane Controlled Trials Register up to May 2014. The weighted mean difference was calculated for net changes in systolic and diastolic BP using fixed-effects or random-effects models. A total of twenty-five eligible studies with 1476 subjects were selected. The analysis showed that the acute intake of tea had no effects on SBP and DBP. However, after long-term tea intake, the pooled mean SBP and DBP were lower by 1.8 and 1.4 mmHg, respectively. When stratified by type of tea, green tea significantly reduced SBP by 2.1 mmHg and decreased DBP by 1.7 mmHg, and black tea showed a reduction in SBP of 1.4 mmHg and a decrease in DBP of 1.1 mmHg. The subgroup analyses showed that the BP-lowering effect was apparent in subjects who consumed tea more than 12 weeks (SBP – 2.6 mmHg and DBP – 2.2 mmHg, both *P* < 0.001). These data suggest that long-term (≥12 weeks) ingestion of tea could result in a significant reduction in SBP and DBP.

Tea is thought to have an anti-hypertension effect in people with elevated blood pressure. Yarmolinsky et al. [24] evaluated the effects of tea on blood pressure in hypertensive individuals. They searched the CENTRAL, PubMed, Embase, and Web of Science databases for relevant studies published from 1946 to September 27, 2013. The selection criteria included: randomized controlled trials of adults with pre-hypertension or hypertension subjected to intervention with green or black tea; controls consisting of placebo, minimal tea intervention, or no intervention; a follow-up period of at least two months. Meta-analyses of 10 trials (834 participants) revealed statistically significant reductions in SBP (2.36 mmHg) and DBP (1.77 mmHg) with tea consumption. Therefore, consumption of green or black tea can reduce blood pressure in individuals within pre-hypertensive and hypertensive ranges, although studies of longer duration and stronger methodological quality are warranted to confirm these findings.

Obesity is known to be one of the most important risk factors for the development of hypertension [25]. Obese individuals have a more than threefold increased likelihood of developing hypertension [26]. Li et al. [27] performed a systematic review and meta-analysis to clarify the efficacy of green tea or green tea extract (GTE) on the blood pressure of overweight and obese adults. They systematically searched electronic databases, conference proceedings including parallel and cross-over randomized controlled trials (RCTs) that examined the effectiveness of green tea or GTE on blood pressure. Data was meta-analyzed using a random effects model, to compare the mean differences in blood pressure changes from baseline in the intervention and placebo groups. Based on the selected criteria, 14 RCTs with a total of 971 participants (47% women) were pooled for analysis. Of the 14 studies, five were conducted in Asia (two Japanese and three Chinese), six in Europe (three in UK, two in Poland, and one in Netherlands), two in USA and one in Australia. All the studies were published between 2006 and 2014. Green tea or GTE produced a significant reduction in both SBP (mean difference of 1.42 mmHg) and DBP (mean difference 1.25 mmHg), compared with the placebo group. Similar results were found in the subgroup and sensitivity analyses. The results demonstrated that green tea or GTE supplementation evokes a small but significant reduction in blood pressure in overweight and obese adults. The data also indicated a strong beneficial effect of green tea or GTE supplementation in this group.

### 2.2. Interventional Trials for General Population

In a long-term follow-up study, Tong et al. [28] recruited 1,109 Chinese men (*n* = 472) and women (*n* = 637) who had participated in the Jiangsu Nutrition Study (JIN). Blood pressure was measured in 2002 and 2007. Tea (green, black and total tea) consumption was quantitatively assessed at the follow-up survey in 2007. Their results showed that total tea and green tea consumption were inversely associated with five-year DBP but not SBP. In the multivariable analysis, those with a daily total tea consumption of at least 10 g had DBP readings 2.41 mmHg (Total) and 3.68 mmHg (green) lower than those who consumed no tea. There was a significant interaction between smoking and total tea/green tea consumption, and diastolic blood pressure change. Green tea consumption was inversely associated with DBP change only in non-smokers and those without central obesity. The authors concluded that the consumption of green tea is inversely associated with five-year blood pressure change in Chinese adults, an effect diminished by smoking. Yang et al. [29] also examined the long-term effects of tea drinking on the risk of hypertension. The study was carefully designed and used a large number of people (1,507 subjects of 711 men and 796 women), and detailed information on tea consumption and other lifestyle and dietary factors associated with hypertension risk. The result showed that those who drank at least 120 mL/day (half a cup) of moderate-strength green or oolong tea for a year, had a 46% lower risk of developing hypertension than the non-tea drinkers. Amongst those who drank 120 to 599 mL/day (two and a half cups), the risk of high blood pressure was reduced by 65%. They concluded that habitual moderate-strength green or oolong tea consumption of at least 120 mL/day for one year significantly reduces the risk of developing hypertension in the Chinese population. Additionally, in a double-blind trial of 111 healthy volunteers, Nantz et al. [30] compared the effects of a standardized capsule containing 200 mg of decaffeinated catechin green tea extract with a placebo. The volunteers consumed a standardized capsule of *Camellia sinensis* compounds twice a day. After three weeks, SBP and DBP was lowered by 5 and 4 mmHg, respectively. After three months, SBP remained significantly lower. 

The relation of tea consumption to cholesterol level and SBP was studied by Stensvold et al. [31], who recruited 9,856 men and 10,233 women (35–49 years of age) from the county of Oppland, Norway. Mean serum cholesterol decreased with an increase in black tea consumption. SBP was inversely related to tea consumption with differences of 2.1 mmHg in men and 3.5 mmHg in women. The data suggested that black tea consumption was associated with a lower systolic blood pressure in Norwegian men and women. Hodgson et al. [32] conducted a randomized controlled trial into the effects of black tea consumption on blood pressure regulation in Australia. A total of 95 men and women aged 35 to 75 years who were regular tea drinkers with a daytime ambulatory SBP between 115 and 150 mmHg, were recruited from the general population. Participants consumed 3 cups/d of regular tea for a four-week run-in period. After 6 months, participants then consumed 3 cups/d of either 1,493 mg powdered black tea solids containing 429 mg of polyphenols and 96 mg of caffeine, or a placebo matched in flavor and caffeine content, but containing no tea solids. The 24 h ambulatory blood pressure was measured at baseline, 3 months, and 6 months. Compared with the placebo, regular ingestion of black tea over 6 months resulted in lower 24 h SBP and DBP. The mean reductions in 24 h SBP were 2.7 mmHg at 3 months and 2.0 mmHg at 6 months. Similarly, the mean reductions in 24 h DBP were 2.3 mmHg at 3 months and 2.1 mmHg at 6 months. Significant differences in blood pressure were also observed for daytime and nighttime measurements, but effects on the overall 24 h blood pressure were mainly driven by daytime blood pressure. The results showed that regular consumption of 3 cups/day of black tea over 6 months, supplying approximately 429 mg/day of polyphenols, resulted in reductions in SBP and DBP of between 2 and 3 mmHg.

Arteries play an important role in cardiovascular function, including abnormalities in blood pressure. Since the aorta has a limited capacity, pressure increases during systole and is partially maintained during diastole by the rebounding of the expanded arterial walls. When arterial stiffness increases, the cushioning function is impaired, leading to a higher SBP and lower DBP. Stiffening of the arterial walls is a very important determinant of the development of hypertension [33,34,35,36]. Therefore, improvement in arterial elasticity is another mechanism for prevention of hypertension. To explore the relationship between habitual tea consumption and arterial stiffness, Lin et al. [37] performed a cross-sectional, epidemiological survey of 6,589 male and female residents aged 40–75 years, in Wuyishan, Fujian Province, China. The results showed that the levels of brachial-ankle pulse wave velocity (ba-PWV) were lowest amongst subjects who consumed tea habitually for more than 10 years, compared with the other 3 subgroups (nonhabitual, 1–5 year, and 6–10 year habitual tea drinkers). In addition, the levels of ba-PWV were lower in subjects who consumed 10–20 and >20 g/day tea habitually, than nonhabitual tea drinkers. As the duration and daily amount of tea consumption increased, the average ba-PWV decreased. Multiple logistic regression models revealed that habitual tea consumption was a positive predictor for ba-PWV. These results indicate that long-term habitual tea consumption may have a protective effect against arterial stiffness.

### 2.3. Interventional Trials for Obese and/or Hypertensive Populations

Nagao et al. [38] ran a double-blind parallel multicenter trial on the effect of green tea extract on body fat and hypertension in 240 Japanese women and men with visceral fat-type obesity. After a two-week diet run-in period, a 12-week trial was undertaken. The subjects ingested green tea containing 583 mg of catechins (catechin group) or 96 mg of catechins (control group) per day. The results showed that a greater decrease in SBP was observed in the catechin group than the control group for subjects whose initial SBP was 130 mmHg or higher. Low-density lipoprotein (LDL) cholesterol also decreased more in the catechin group. No adverse effect was found. Brown et al. [39] also ran a randomized controlled trial to investigate the effect of dietary supplementation with EGCG on insulin resistance, and associated metabolic risk factors including high blood pressure, in obese man. Overweight or obese male subjects, aged 40–65 years, were randomly assigned to take 400 mg capsules of EGCG (*n* = 46) or the placebo lactose (*n* = 42), twice a day for eight weeks. Oral glucose tolerance testing and measurement of metabolic risk factors (BMI, waist circumference, percentage body fat, blood pressure, total cholesterol, LDL-cholesterol and HDL-cholesterol) were performed pre- and post-intervention. The results showed that EGCG treatment had no effect on insulin sensitivity, insulin secretion or glucose tolerance, but did reduce DBP (mean change: placebo −0·058 mmHg; EGCG -2·68 mmHg). Significant changes in the other metabolic risk factors were not observed. EGCG treatment also had a positive effect on the mood of the participant. Recently, Nogueira et al. [40] ran a crossover randomized clinical trial investigating the short-term effects of green tea on blood pressure and endothelial function in obese pre-hypertensive women. Participants were randomly allocated to receive daily three capsules containing either 500 mg of GTE or a matching placebo for four weeks, with a washout period of two weeks between treatments. After four weeks of GTE supplementation, there was a significant decrease in SBP in comparison with the placebo over 24 h (3.61 vs. 1.05 mmHg), in the daytime (3.61 vs. 0.80 mmHg), and at night (3.94 vs. 1.90 mmHg). Differences in DBP and all other parameters were not significant between the GTE and placebo groups. The data suggests that in obese pre-hypertensive women, short-term daily intake of GTE may decrease blood pressure. Bogdanski et al. [41] examined the effects of GTE with insulin resistance and associated cardiovascular risk factors in obese, hypertensive patients. In this double-blind, placebo-controlled trial, 56 obese, hypertensive subjects were randomized to receive a daily supplement of 1 capsule that contained either 379 mg of GTE or a matching placebo, for 3 months. At baseline and after 3 months of treatment, the anthropometric parameters, blood pressure, plasma lipid levels, glucose levels, creatinine levels, and insulin levels were assessed. After three months of supplementation, both SBP and DBP had significantly decreased in the GTE group as compared with the placebo group (*P* < 0.01). 

Black tea accounts for 78% of the world’s tea production and is consumed worldwide. Therefore, it is important to determine whether black tea has an anti-hypertension effect. Grassi et al. [42] investigated the effect of black tea on blood pressure and vessel wave reflections before and after fat consumption in hypertensive patients. In a randomized, double-blind, controlled, cross-over study, 19 patients were assigned to consume black tea (129 mg flavonoids) or a placebo twice a day for eight days (13-day wash-out period). Digital volume pulse and BP were measured before and 1, 2, 3 and 4 h after tea consumption. Measurements were performed in a fasted state and after a fat load. The authors found that fat consumption led to increase wave reflection, which was counteracted by tea. The results indicate that black tea consumption decreases SBP and DBP by 3.2 mmHg and 2.6 mmHg, respectively, and prevented blood pressure increase after a fat consumption. These findings indicate that regular consumption of black tea may play an important role in cardiovascular protection.

### 2.4. Interventional Trials for Diabetic Populations

High blood pressure is more common in people with diabetes. It was estimated that two-thirds of patients with type 2 diabetes have high blood pressure [43]. Mozaffari et al. [44] conducted a randomized clinical trial in which 100 mildly (equal to Stage 1 hypertension of new guideline) hypertensive patients with diabetes were randomly assigned into a green tea treatment group. The patients were instructed to drink green tea infusion three times a day, 2 h after each meal for four weeks. Blood pressure was measured at days one and 15, and at the end of the study. The results showed that the SBP of the green tea group was lower at the end of the study (from 119.4 to 114.8 mmHg). The DBP was also lower by the end of the study (from 78.9 to 75.3 mmHg). The therapeutic effectiveness of tea by the end of the intervention was 39.6% in the green tea group. This data indicates that mildly hypertensive type 2 diabetic individuals who drink green tea daily show significantly lower SBP and DBP. Another study [45] was conducted in Japan with 60 volunteers who had fasting blood glucose levels of ≥6.1 mmol/L or non-fasting blood glucose levels of ≥7.8 mmol/L. The intervention group consumed a packet of GTE containing 544 mg polyphenols (456 mg catechins) daily for the first two months, and then entered a two-month nonintervention period. Supplementation of GTE powder led to a significant reduction in DBP, but no significant changes in SBP.

### 2.5. Intervention Trials for Aging Populations

Hypertension generally increases with age [34]. Hodgson et al. [32] evaluated the effect of long-term regular ingestion of tea on blood pressure in older women. A total of 218 women over 70 years of age was included in this cross-sectional study. The results indicate that tea intake is associated with significantly lower SBP and DBP. Furthermore, a 250 mL/day (one cup) increase in tea intake was associated with a 2.2 mmHg lower SBP and a 0.9 mmHg lower DBP. This suggests that regular tea consumption may have a favorable effect on blood pressure in older women. Recently, Yin et al. conducted a cross-sectional study of blood pressure and tea consumption an elderly population in Jiangsu, China [35]. A total of 4579 older adults aged 60 years or older participated in this study. And a linear regression model was applied for analysis of association between tea consumption and risk of hypertension. The results showed that higher tea consumption frequency was found to be associated with lower systolic BP values, after adjusting for the effect of age, sex, education level, lifestyle-related factors, and cardiometabolic confounding factors in overall (*P* = 0.0003), normotensive (*P* = 0.017) and participants without anti-hypertensive treatment (*P* = 0.027). Significant inverse association between diastolic BP and frequency of tea consumption was also observed in the overall subjects (*P* = 0.003). In multivariate logistic analyses, habitual tea drinking was inversely associated with presence of hypertension (*P* = 0.011).

The main clinical studies of blood pressure-lowering effects by tea in humans are summarized in Table 1. 

## 3. Tea Metabolites Regulating Blood Pressure in Animal Studies

A growing number of reports indicate that tea has beneficial effects on blood pressure in various animal models. As early as 1984, Henry et al. [46] investigated the effect of decaffeinated tea on chronic psychosocial hypertension in CBA mice. They found that tea polyphenols (not caffeine) reduced blood pressure from 150 to 133 mmHg. Negishi et al. [47] evaluated the hypotensive effect of black and green tea polyphenols by using a stroke-prone spontaneously hypertensive (SHR) rat model. The male rats were divided into three groups: the control group consumed tap water (30 mL/d); the black tea polyphenol group (BTP) consumed water containing 3.5g/L thearubigins, 0.6 g/L theaflavins, 0.5 g/L flavonols and 0.4 g/L catechins; and the green tea polyphenol group (GTP) consumed water containing 3.5 g/L catechins, 0.5 g/L flavonols and 1 g/L polymetric flavonoids. The telemetry system was used to measure blood pressure, which was recorded continuously every 5 min for 24 h. During the daytime, SBP and DBP were significantly lower in the BTP and GTP groups than in the controls. As the amounts of polyphenols used in this experiment correspond to daily consumption of tea consumers, regular consumption of black and green tea may provide some protection against hypertension in humans.

EGCG is a polyphenol that makes up approximate 30% of the solids in green tea [48]. Potenza et al. [49] studied the effect of EGCG treatment on cardiovascular and metabolic function using a SHR rat model. In acute studies, EGCG (1–100 µM) elicited dose-dependent vasodilation in mesenteric vascular beds (MVB) isolated (ex vivo) from SHR. In chronic studies, nine-week-old SHR were treated by gavage for 3 weeks with EGCG (200 mg/kg/day), enalapril (30 mg/kg/day), or vehicle. They found that both EGCG and enalapril therapy significantly lowered SBP in the SHR rat. Using the Goto-Kakizaki rat model, Igarashi et al. [50] also demonstrated that a diet containing 0.2% tea catechins tended to maintain SBP (in the latter stages of a 76-day feeding period) at lower levels than in subjects not receiving dietary catechins.

Tea plant has a characterized amino acid, theanine. Yasuhiko et al. [51] investigated the effect of green tea, rich inγaminobutyric acid, on blood pressure in young and old Dahl salt-sensitive rats. For therapeutic effect, 11-month-old rats were fed a 4% NaC1 diet for 3 weeks, then were given water (group W), an ordinary tea solution (group T), or a GABA-rich tea solution (group G) for 4 weeks. After this treatment, blood pressure was significantly decreased in group G (176 mmHg) compared with group W (207 mmHg) or group T (193mmHg). For the preventive experiment, 5-week-old rats were fed a 4% NaCI diet and divided into groups W, T, and G. After 4 weeks of treatment, although blood pressure was comparable in groups W and T (165 vs. 164 mmHg), it was significantly lower in group G (142 mmHg). Therefore, GABA-rich tea seems to decrease the established high blood pressure and prevent the development of hypertension in Dahl rats fed a high salt diet. Yokogoshi et al. [52] also studied the hypotensive effect of γ-glutamylmethylamide glutamic acid (GMA) and theanine in SHR rats. Glutamic acid (2000 mg/kg) did not alter the rat blood pressure but the same dose of theanine decreased rat blood pressure significantly. GMA administration to SHR also reduced the blood pressure significantly, and its hypotensive action was more effective than that of theanine only administration.

Tannic acid is a water-soluble polyphenol that is present in tea. Turgut et al. [53] evaluated the effect of tannic acid on SBP in a L-NNA-induced essential hypertension rat model. Tannic acid was intraperitoneally injected at a dose of 50 mg/kg for 15 days. Compared with the hypertension group, tannic acid administration significantly decreased blood pressure values after 20 and 30 days. Sagesaka-Mitane et al. [54] investigated the effect of tea-leaf saponin on blood pressure using SHR rats. Tea-leaf saponin led to a time- and dose-dependent reduction in blood pressure when it was administered orally to young SHR (7 weeks old) for 5 days. Oral administration of tea-leaf saponin (100 mg/kg) to older SHR (15 weeks old) for 5 days decreased the mean blood pressure by 29.2 mmHg, compared to the control group. Single administration of tea-leaf saponin at 50 mg/kg showed a long-lasting hypotensive effect which was as potent as that of enalapril maleate at 3 mg/kg. Their data proved that both tannic acid and tea-leaf saponin have the effect of lowering blood pressure in hypertensive rats.

## 4. Molecular Mechanisms of Tea Regulating Blood Pressure 

A growing body of evidence indicates that oxidative stress and the inactivation of NO by vascular superoxide anion play critical roles in the development of hypertension [55]. Vascular superoxide anion is enhanced in angiotensin II (Ang II)-induced hypertension, mainly attributable to nicotinamide adenine dinucleotide phosphate (NADPH) oxidase activation by Ang II [56,57]. In addition, an excess of vascular superoxide anion production has also been found in spontaneously hypertensive and deoxycorticosterone acetate (DOCA) salt hypertension [58], and in mineralocorticoid hypertensive rats [59]. Antonello et al. [60] investigated the preventative effects of GTE on Ang II-induced blood pressure increases using the Sprague-Dawley (SD) rat model. Male SD rats were randomly assigned to drinking water with or without GTE (6 mg/mL) and received (by osmotic mini-pumps) a vehicle, high (700 μg/kg/day) or low (350 μg/kg/day) dose of Ang II for 13 days. Night-time and daytime SBP and DBP were recorded with telemetry. By day two of infusion, the AngII group showed significantly higher SBP and DBP during the day and night, compared to all other groups. Moreover, GTE significantly lowered both SBP and DBP throughout the study period, compared with the AngII group. In addition, GTE blunted the increase in HO-1, p22phox, and SOD-1 mRNA in the aorta caused by Ang II. The data suggests that GTE prevented hypertension induced by a high Ang II dose, possible by the prevention or scavenging of superoxide anion generation. GTE consists of both catechins and caffeine. In order to remove the caffeine contribution, Ihm et al. [61] studied the effect of decaffeinated GTE on hypertension and insulin resistance in an Otsuka Long-Evans Tokushima Fatty (OLETF) rat model of metabolic syndrome (MetS). OLETF rats were randomized into a saline-treated group and a group treated with decaffeinated-GTE (25 mg/kg/day). They found that decaffeinated-GTE significantly reduced BP (130 vs. 121 mmHg). In addition, decaffeinated-GTE significantly reduced vascular reactive oxygen species (ROS) formation and NADPH oxidase activity, and improved endothelium-dependent relaxation in the thoracic aorta of OLETF rats. Decaffeinated-GTE also suppressed the expression of p47 and p22phox in the immunohistochemical staining and stimulated phosphorylation of endothelial nitric oxide synthase (eNOS) and Akt in immunoblotting of the aortas. These results revealed that decaffeinated-GTE reduced the formation of ROS and NADPH oxidase activity and stimulated phosphorylation of eNOS and Akt in the aorta of a rat model of MetS, which resulted in improved endothelial function and eventually in lower blood pressure.

Gómez-Guzmán et al. [62] studied the effects of chronic treatment with epicatechin on blood pressure, endothelial function, and oxidative status using the DOCA-salt-induced hypertensive rat model. The rats were treated for 5 weeks with (-)-epicatechin at 2 or 10 mg/kg/day. The higher dose of epicatechin prevented the increase in SBP induced by DOCA-salt. The authors found that aortic superoxide levels were elevated in the DOCA-salt group and abolished by both doses of epicatechin. However, only epicatechin at 10 mg/kg/day reduced the DOCA-salt-induced increase in aortic NADPH oxidase activity, and p47phox and p22phox gene overexpression. They also showed that epicatechin increased the transcription of nuclear factor-E2-related factor-2 (Nrf2) and Nrf2 target genes in the aortas of control rats. Furthermore, epicatechin improved the impaired endothelium-dependent relaxation response to acetylcholine and increased the phosphorylation of both Akt and eNOS in aortic rings. Epicatechin also induced a decrease in ET-1 release, systemic and vascular oxidative stress, and inhibition of NADPH oxidase activity. Galleano et al. [63] used a SHR rat to investigate the effects of dietary consumption of (-)-epicatechin on blood pressure regulation. They found that consumption of 0.3% (-)-epicatechin for 2 and 6 days significantly deceased SBP by 27 and 23 mmHg, respectively. They also studied the mechanism relating to decreased blood pressure and observed a 173% increase in nitric synthase (NOS) activity in the aorta of (-)-epicatechin SHR on day six, compared with non-supplemented SHR. These findings infer that (-)-epicatechin can modulate blood pressure in hypertensive rats by increasing NO levels in the vasculature. Litterio et al. [64] also investigated the effects of (-)-epicatechin on blood pressure in No-nitro-L-arginine methyl ester (L-NAME)-treated rats. The administration of (-)-epicatechin prevented the 42 mmHg increase in blood pressure associated with the inhibition of NO production in a dose-dependent manner (0.2–4.0 g/kg diet). This blood pressure effect was associated with a reduction in L-NAME-mediated increases in the indexes of oxidative stress, and with a restoration of the NO concentration. At the vascular level, none of the treatments modified NOS expression, but (-)-epicatechin administration alleviated the L-NAME-mediated decrease in eNOS activity and increase in both superoxide anion production and NOX subunit p47phox expression. In summary, (-)-epicatechin prevented the increase in blood pressure and oxidative stress and restored NO bioavailability. 

It is well known that sympathetic nerve activity plays a pivotal role in blood pressure regulation. Tanida et al. [65] reported the effects of oolong tea (OT) on renal sympathetic nerve activity (RSNA) and spontaneous hypertension in SHR rats. They found that intraduodenal injection of OT in urethane-anesthetized rats suppressed RSNA and decreased blood pressure. In addition, pretreatment with the histaminergic H3-receptor-antagonist thioperamide or bilateral subdiaphragmatic vagotomy eliminated the effects of OT on RSNA and blood pressure. Furthermore, drinking OT for 14 weeks reduced blood pressure elevation in SHR rats. These results suggest that OT may exert its hypotensive action through changes in autonomic neurotransmission via an afferent neural mechanism. The authors also found that intraduodenal injections of decaffeinated OT lowered RSNA and blood pressure to the same degree as caffeinated OT, indicating that substances other than caffeine may function as effective modulators of RSNA and blood pressure. Han et al. [66] provided additional evidence that EGCG counteracts caffeine-induced increases in arterial pressure, adrenaline and noradrenaline levels in the blood, and heart rate. The authors suggested that EGCG may exhibit these properties by decreasing the levels of catecholamines in the blood. 

The stimulatory effects of caffeine may be reduced by the amount of EGCG in green tea. Recently, Garcia et al. [67] investigated the effects of green tea on blood pressure and sympathoexcitation in a L-NAME-induced hypertensive rat model. They found that L-NAME-treated rats exhibited an increase in blood pressure (165 mmHg) compared with control rats (103 mmHg), and that green tea-treatment reduced hypertension (119 mmHg). Hypertensive rats showed a higher renal sympathetic nerve activity (161± 12 spikes/second) than the control group (97 ± 2 spikes/second); green tea also decreased this parameter in the hypertensive treated group (125 ± 5 spikes/second). Arterial baroreceptor function and vascular and systemic oxidative stress were improved in hypertensive rats after green tea treatment. Taken together, short-term green tea treatment improved cardiovascular function in a hypertension model characterized by sympathoexcitation, possible due to its antioxidant properties.

Blood vessels are able to self-regulate tone and adjust blood flow in response to changes to the local environment, due to their capacity to respond to physical and chemical stimuli in the lumen. Endothelium is an inner layer of blood vessels, which directly sense the physical and chemical stimuli. Endothelial-dependant vasodilation contributes to the maintenance of an adequate blood flow to cells and tissues. Therefore, the endothelium plays a key role in the control of vascular tone by releasing several vasorelaxing factors including nitric oxide (NO) and endothelium-derived hyperpolarizing factor (EDHF). The calcium signal is a pivotal pathway leading to the activation of eNOS. The phosphatidylinositol 3-kinase/Akt (PI3-kinase/Akt) pathway is another significant signal pathway in the activation of eNOS. 

Anter et al. [68] found that when porcine aortic endothelial cells are exposed to components of black tea, the polyphenol fraction acutely enhanced nitric oxide bioactivity. This effect involved eNOS phosphorylation at Ser-1177 (activator site) and dephosphorylation at Thr-495 (an inhibitor site), consistent with increased eNOS activity. Furthermore, they demonstrated that black tea polyphenol-induced eNOS activation was dependent upon the PI3-kinase/Akt pathway. These stimulatory effects were found to be calcium-dependent, and involved both intracellular and extracellular calcium, and the p38 mitogen-activated protein kinase (p38 MAPK) upstream of the PI3-kinase/Akt pathway. Caveolin-1 is a major negative regulator of eNOS activity. Green tea polyphenols down-regulate caveolin-1 gene expression in a time- and dose-dependent manner, via the activation of extracellular signal-regulated kinase 1/2 (ERK 1/2), and inhibition of p38 MAPK signaling pathways, in bovine aortic endothelial cells (thereby increasing eNOS activation) [69]. However, using Ca2+-deprived endothelial cells, Ramirez-Sanchez et al. [70] found that (-)-epicatechin induced calcium-independent eNOS activation. The results demonstrated that (-)-epicatechin induced a partial AKT/HSP90 migration from the cytoplasm to the caveolar membrane fraction where HSP90, AKT, and eNOS physically associated. Thus, under Ca2+ free conditions, (-)-epicatechin stimulates NO synthesis via the formation of an active complex between eNOS, AKT, and HSP90. Moreover, Fyn (a member of the Src family), mediates the EGCG -induced PI3-kinase/Akt-mediated activation of eNOS [71]. In addition, another study showed that transient receptor potential vanilloid type 1(TRPV1) is pivotal for EGCG-mediated activation of eNOS [72]. The results indicate that EGCG may trigger activation of TRPV1-Ca2+ signaling, which leads to phosphorylation of Akt, AMPK, and CaMKII, and further eNOS activation and NO production.

Vascular smooth muscle is a main component of vascular vessels. The contraction and relaxation of vascular smooth muscle determine vascular tone, and play a pivotal role in the regulation of blood pressure. The effects of GTE on arterial blood pressure and contractile responses of isolated aortic strips were assessed in normotensive rats [73]. Phenylephrine -induced contractile responses were significantly inhibited in the presence of GTE (0.3–1.2 mg/mL) in a dose-dependent fashion. Also, high potassium-induced contractile responses were depressed in the presence of 0.6–1.2 mg/mL of GTE, but not affected in low concentrations (0.3 mg/mL). Interestingly, the infusion of a moderate dose of GTE (10 mg/kg/30 min) caused a significant reduction in pressor responses induced by intravenous norepinephrine, although EGCG (1 mg/kg/30 min) did not affect them. The authors further demonstrated that dose-dependent depressor action of GTE is (at least) partly due to the inhibition of adrenergic αl-receptors. GTE also promoted a relaxation in the isolated aortic strips of rats via the blockade of adrenergic α1-receptors. These findings suggest that there is a big difference in the vascular effect of GTE and EGCG. 

Renin also plays a pivotal role in the development of hypertension. Patients with low renin (i.e., salt-sensitive hypertension) represent approximately 30% of the essential hypertensives, and show a poor therapeutic response to angiotensin-converting enzyme inhibitors and angiotensin receptor blockers [74]. Renin inhibitory activities of three tea products have been investigated [75] and strong inhibition was observed with water extracts from fermented oolong and black tea. The authors demonstrated that theasinensin B, theasinensin C, strictinin, and a hexose sulfate with a galloyl moiety, exhibited IC50 values (against renin activity) of 19.33, 40.21, 311.09 and 50.16 μM, respectively. Moreover, the potent inhibitor theasinensin B was present only in black tea, and monomeric catechins did not contribute significantly to the renin inhibitory activities. These results suggest another potential pathway through which tea consumption helps to control hypertension.

Hypertension may be caused by vascular inflammation and remodeling [76,77]. Inflammation may play a role in the pathogenesis of hypertension or it may characterize a functional state of the vessel wall due to high blood pressure. IL-6 (a proinflammatory molecule) and MMP enzymes have important functions in vessel remodeling in vasculature. Mahajan et al. [78] demonstrated significant induction of IL-6 and MMP-9 expression in THP-1 macrophages by normocholesterolaemic hypertensive sera. Also, green tea polyphenols were found to significantly attenuate this induced expression in vitro. Their study revealed the existence of a potential causal relationship between hypertension, inflammation and vascular remodeling. Monocyte chemotactic protein-1 (MCP-1) plays a pivotal role in the recruitment of monocytes and amplifies inflammatory responses. Ahn et al. [79] studied the mechanisms by which EGCG inhibits tumor necrosis factor-α (TNF-α)-induced MCP-1 production in bovine coronary artery endothelial cells. The data showed that EGCG inhibited TNF-α-induced MCP-1 production, but it blunted Akt phosphorylation and TNF-α activation of TNFR1, which subsequently resulted in reduced MCP-1 production. 

Endothelin-1 (ET-1) is the most potent vasoconstrictor produced in the blood vessel wall and has been shown to contribute to the pathogenesis of salt-sensitive hypertension in animals and humans [80]. ET-1 also augments vascular superoxide anion production, at least in part, via the ETA/NADPH oxidase pathway, leading to endothelial dysfunction and hypertension [81,82]. Nicholson et al. [83] reported that EGCG inhibited both eNOS and ET-1 mRNA expression at the physiological concentration (0.1 μM). These observed effects on gene expression should result in vasodilation and subsequent reduction in blood pressure. Reiter et al. [84] found that the phosphatidylinositol 3-kinase-dependent transcription factor FOXO1 mediates the effects of EGCG in the regulation of ET-1 expression in endothelial cells. EGCG treatment (10 μM for 8 h) of human aortic endothelial cells reduced expression of ET-1 mRNA, protein, and ET-1 secretion. A further study indicated that EGCG decreases ET-1 expression and secretion from endothelial cells, partly via Akt- and AMPK-stimulated FOXO1 regulation of the ET-1 promoter. This finding highlights another potential mechanism for the inhibition of vascular ET-1 release, due to ECGC in tea, which facilitates the protection of endothelial function and lowering of blood pressure.

The underlying mechanisms of tea regulating blood pressure using cell culture and animal models are illustrated in the Figure 1.

## 5. Discussion and Prospective

In conclusion, the bulk of evidence suggests that consumption of both green and black tea is associated with reductions in blood pressure, despite the negative results of some studies [85,86]. Many factors may influence the effect of tea consumption on blood pressure in human population studies. For example, the duration of tea consumption has an important impact on blood pressure. In a cohort of Norwegian men and women, higher consumption of black tea was associated with lower SBP [26]. However, in a four-week randomized, controlled, crossover trial in normotensive men and women, drinking six mugs of tea daily had no significant effect on clinic-measured blood pressure [87]. Other human intervention studies investigating the short-term effects of tea consumption on blood pressure also failed to achieve a positive outcome [88,89]. Differences in the origin of the tea and its secondary metabolites are further variables that occur between studies. In addition to flavonoids, tea contains caffeine, which causes a short-term increase in blood pressure [90]. These increases in blood pressure should be considered in the design of research projects. Hodgson [91] observed an increase in blood pressure 30 min after the ingestion of green or black tea in normotensive men; this increase was not evident after 60 min. Interestingly, the increase in blood pressure was greater than that induced by an equivalent dose of caffeine alone, suggesting that tea or tea polyphenols may also promote acute increases in blood pressure. Consumption of either black or green tea for seven days had no effect on 24 h ambulatory blood pressure in the same population. The same study showed that the acute hypertensive effect of tea consumption was blunted when tea was consumed with food [30]. The initial blood pressure is another factor that should be considered. Some case studies examined normotensive populations and those already well-controlled by anti-hypertensive therapy. In this situation, it is difficult to demonstrate a blood pressure lowering effect of tea intervention. Further large-scale investigations with longer terms of observation and tighter controls, are needed to determine optimal doses in subjects with varying degrees of hypertensive risk factors, and to determine differences in beneficial effects amongst diverse populations. Moreover, data from tissues and cell cultural studies have shown that tea and its secondary metabolites have important roles in relaxing smooth muscle contraction, enhancing eNOS activity, reducing vascular inflammation, inhibiting rennin and ET-1 activity and anti-vascular oxidative stress. However, the exact molecular mechanisms of these activities remain to be elucidated.

According to previous research, hypertension is more common in diabetic patients. Hypertension is present in more than 50% of patients with diabetes mellitus [92]. Recently, a systematic review and meta-analysis of randomized controlled trials involved in 27 studies (1898 participants) suggested that green tea may reduce fasting blood glucose (FBG) levels compared with placebo/water. Subgroup analysis showed that the effect of green tea on fasting blood glucose levels was significant only in studies with a mean age of < 55-years-old or Asian-based studies [93]. Previous researches reported that the potential beneficial effect of green tea on glucose metabolism may be mediated by EGCG, the most abundant catechin present in green tea [94]. Waltner-Law et al. reported that EGCG reduces hepatic glucose production by increasing tyrosine phosphorylation of the insulin receptor and insulin receptor substrate-1 in H4IIE rat hepatoma cell [95]. Recent studies have also suggested that green tea and yellow tea increase insulin sensitivity and glucose metabolism, preventing progress of type 2 diabetes [96]. Ortsäter et al. also reported that EGCG preserves islet structure and enhances glucose tolerance in db/db mice [97]. Therefore, teas decrease diabetes-related hypertension might mediate by ameliorating diabetes complication, which prevents high glucose induced damages of vascular endothelial and smooth muscle cells and maintain normal dilation and contraction of vascular system. The exact mechanisms of teas preventing diabetes-related hypertension need further investigation.

Clinical data shows that a 5-mmHg blood pressure reduction can reduce the risk of stroke and ischemic heart disease by 34 and 21%, respectively [6,7]. Although no clinical data available to demonstrate CVD outcome of 1-3 mmHg reduction of SBP and DBP, such small reductions of BP may benefit Stage 1 hypertension patients to control BP at range. Based on animal models, teas treatment significantly decreased SBP and DBP. There is a huge disparity between blood pressure lowering effects of tea in animal studies and human intervention. This difference may due to simplicity of animal studies and complicity of human intervention. In animal experiment, genetic background (same strain), living environment (standard facility), dosage and duration of treatment can be strictly controlled. Therefore, it is easy to see BP lowing effect by teas. For human population studies, complicated social and genetic backgrounds, varying degrees of hypertensive, dietary intaking and physical activity all can affect BP lowing effect of teas. Therefore, tighter controls are needed to determine BP lowing effects of tea in human populations.

## 6. Conclusions

In summary, human interventions and animal studies have confirmed that both tea and tea metabolites have anti-hypertensive effects although some controversial reports existed. The underlying mechanisms include relaxing smooth muscle contraction, enhancing endothelial nitric oxide synthase activity, reducing vascular inflammation, inhibiting rennin activity, and anti-vascular oxidative stress based on ex-vivo tissue and in vitro cell culture studies.

## Figures and Tables

**Figure 1 nutrients-11-01115-f001:**
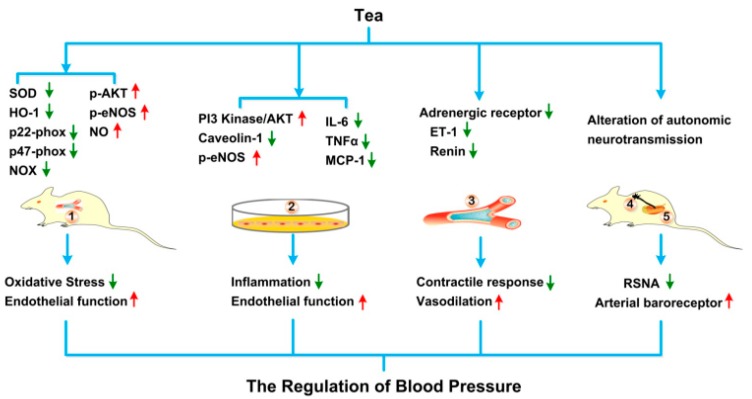
The underlying mechanisms of tea regulating blood pressure using animal, tissue and cell line models. After intake or adding into cell cultures, tea and its bioactive ingredients can alter several blood pressure regulating processes, including ① alleviation of the oxidative stress and improvement of the endothelial function in the aortas in vivo; ② mitigation of the inflammation and amelioration of the endothelial function in the aortic epithelial cell lines in vitro; ③ suppression of contractile response and improvement of vasodilation in the aortic tissues in vitro; ④,⑤ inhibition of renal sympathetic nerve activity and amelioration of arterial baroreceptor. Abbreviation: SOD, superoxide dismutase; HO-1, heme oxygenase 1; NOX, NADPH oxidase; p-AKT, phosphorylated protein kinase B; p-eNOS, phosphorylated endothelial nitric oxide synthase; NO, nitric oxide; PI3 Kinase, phosphatidylinositol 3 kinase; IL6, Interleukin 6; TNFα, Tumor necrosis factor α; MCP-1, Monocyte chemotactic protein-1; ET-1, Endothelin-1; RSNA, renal sympathetic nerve activity.

**Table 1 nutrients-11-01115-t001:** Blood pressure-lowering effects of tea in human interventions.

No.	Methods of Study	Selected Participants	Tea or Dosage & Duration	Test Site	Primary Outcomes and Comments	Year (Citation)
1	Meta-analysis	13 studies	Green tea	Australia	Significantly reduced SBP by 2.08 mmHg and DBP by 1.71 mmHg. Good methodology analysis and trusted results	2014 [20]
2	Meta-analysis	13 randomized controlled trials across, 1367 subjects	Green tea polyphenols (<582.8 mg/day), ≥12 weeks	Several countries	Significantly reduced SBP level by 1.98 mmHg and DBP by 1.92 mmHg. Good methodology analysis, a large population and trusted results	2014 [21]
3	Meta-analysis	11 randomized controlled trials, 378 subjects	4–5 cups of black tea	Netherland	The SBP and DBP were decreased by 1.8 mmHg and 1.3 mmHg. Good methodology, and trusted results	2014 [22]
4	Meta-analysis	25 eligible studies, 1476 subjects	Both green and black tea intake,≥ 12 weeks	USA	Long-term (≥12 weeks) ingestion of tea could result in a significant Reduction in SBP and DBP Good methodology, a large population analysis and trusted results	2014 [23]
5	Meta-analysis	10 trials (834 participants) hypertensive individuals	Tea regular consumption	UK	Significant reductions in SBP (2.36 mmHg) and DBP (1.77 mmHg). Good methodology and trusted results.	2015 [24]
6	Meta-analysis	971 overweight and obese adult participants (47% women)	Green tea or green tea extract	China	Significant reduction in both SBP (1.42 mmHg) and DBP (1.25 mmHg). Good methodology and trusted results.	2015 [27]
7	Multivariable analysis	472 men and 637 women	Tea (green, black and mixed teas) consumption	China	The consumption of green tea is inversely associated with five-year BP change in Chinese adults, an effect was diminished by smoking. Long term study and trusted results.	2014 [28]
8	Meta-analysis	711 men and 796 women	120 mL/day (half a cup)green or oolong tea, a year	China	Significantly reduces the risk of developing hypertension. A large special population, and trusted results.	2004 [29]
9	Meta-analysis	9,856 men and 10,233 women (35–49 years of age)	Black tea regular consumption	Norway	SBP was inversely related to tea consumption with differences of 2.1 mmHg in men and 3.5 mmHg in women. Good methodology, a large population analysis and trusted results.	1992 [31]
10	Randomized controlled trial	95 men and women aged 35 to 75 who were regular tea drinkers	3 cups/d of regular black tea consumption, ≥ 6 months	Australia	Reductions in SBP and DBP of between 2 and 3 mmHg. A small population trial, and a reasonable result.	2012 [32]
11	Double-blind parallel multicenter trial	240 Japanese women and men with visceral fat-type obesity	583 mg of catechins or 96 mg of catechins per day, green tea	Japan	Catechin group decreased initial SBP that is 130 mmHg or higher. A small population trial, and a reasonable result.	2007 [38]
12	Randomized controlled trial	Overweight or obese male subjects, aged 40–65 years	400 mg capsules of EGCG (*n* = 46) or the placebo lactose (*n* = 42), twice a day for eight weeks.	UK	EGCG treatment did reduce DBP (mean change: placebo -0·058 mmHg; EGCG -2·68 mmHg). A small population trial, and a reasonable result.	2009 [39]
13	Randomized clinical trial	Obese pre-hypertensive women	500 mg of GTE or a matching placebo consumption, four weeks	Brazil	Short-term daily intake of GTE may decrease BP in obese pre-hypertensive women. A small population, short term trial, and a reasonable result.	2016 [40]
14	Double-blind	56 obese, hypertensive subjects	Daily supplement 379 mg of GTE or a matching placebo, 3 months	Poland	Both SBP and DBP had significantly decreased compared with the placebo group. A small population trial, and a reasonable result.	2012 [41]
15	Randomized, double-blind	19 patients	Black tea (129 mg flavonoids) or a placebo twice a day for eight days.	Italy	Black tea consumption decreases SBP and DBP by 3.2 mmHg and 2.6 mmHg, respectively, and prevented BP increase after a fat consumption. A small population trial may results in bias result.	2015 [42]
16	Randomized clinical trial	100 stage1 hypertensive patients with diabetes	Drink green tea infusion three times a day 2 hours after each meal, four weeks	Iran	Stage1 hypertensive type 2 diabetic individuals who drink green tea daily show significantly lower SBP and DBP. A small population trial, and a reasonable result.	2013 [44]
17	Cross-sectional study	60 volunteers, fasting blood glucose levels of ≥6.1 mmol/L or non-fasting blood glucose levels of ≥7.8 mmol/L	544 mg polyphenols (456 mg catechins) daily consumption, 3 months	Japan	Supplementation of GTE powder led to a significant reduction in DBP, but no significant changes in SBP. A small population trial, and a reasonable result.	2008 [45]
18	Cross-sectional study	218 women over 70 years old	250 mL/day (one cup)	Australia	Regular tea consumption may have a favorable effect on BP in older women. A small population trial, and a reasonable result.	2003 [32]
19	Cross-sectional study	4579 adults aged 60 years or older	tea consumption questionnaire	China	Higher tea consumption frequency was associated with lower systolic BP. A large population trial, and a reasonable result.	2017 [35]

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
