# Peer review of "Effects and Mechanisms of Tea Regulating Blood Pressure: Evidences and Promises"

_nutrients, 2019, doi:10.3390/nu11051115_

Round 1
Reviewer 1 Report
Tea is one of the most popular drinks worldwide and there is great interest in understanding effects of tea on blood pressure as well as on our general well-being. The review article by Li and colleagues attempted to summarize anti-hypertensive effects of different types of teas along with discussions of the pharmacological basis of such effects. My comments are as follows:
1. Overall structure of the review is poor. It does not talk anything about the methodology or analysis methods or controls. It is not clear why it was important to discuss “Main types of teas” under a separate headline, where the authors have talked a great deal about the classification of tea and their production methods. In addition, separating and discussing animal studies (headline 4, line 311) from those on tissues and cells (headline 5, line 442) has interrupted the natural flow of the discussion.
2. Definition of hypertension is problematic. The authors did not specify what guidelines were followed for defining hypertension. The latest guidelines for hypertension published in November 2017 by ACC/AHA replaces previous guidelines. Mean BP is not consistent with the values for hypertension (e.g. heading 3.4). Use of term such as ‘mildly hypertensive’ is difficult to understand, as it does not have any numbers to specify this condition. If hypertension is more common in diabetic patients (as mentioned in 3.4) and tea does not have any effects on glucose tolerance, insulin secretion etc., how does it lower BP in this patients? Relevant discussion and commentary are missing throughout. Human intervention studies show only 1-3 mmHg reduction of SBP and DBP. However, data and relevant commentary is missing as to how such small reductions of BP lowers CV risk. There is a huge disparity between blood pressure lowering effects of tea in animal studies and human studies and, it is expected that the authors shed light on such observed differences.
3. The authors only described the anti-hypertensive effects of tea without any mention about other studies where tea has been shown to increase blood pressure. The review is one-sided with no consideration for the broader effects tea may have on human health and blood pressure.
Minor comments:
4. The title of the article does not convey a clear message and therefore looks less attractive at first sight.
5. Relevant reference is missing in many places. For example:
In page 1, line 36, ‘CVDs have overtaken cancer as the major cause of death……..”.
In page 1, line 38-39, ‘ Of these risk factors, high blood pressure is the most dangerous factor linked to CVDs death events.
There are so many places in the manuscript where appropriate references are missing. Also, references cited do not cover the major studies supporting that claim. For example:
In page 2, lines 46 – 50, there are more than just one reference, which should have been cited. These are just examples; the manuscript should be thoroughly revised for consistency.
6. The authors presented a table for the human studies but it is missing some key information that the readers would be interested in. For example, it does not mention experiment design or conditions used for the studies or for analysis methods. There is no conclusion/comments either.
Author Response
Dear Editors and Reviewers:
Thank you very much for your letter and the referees’ reports! Based on your comments and suggestions, our manuscript entitled, “Effects and Mechanisms of Tea Regulating Blood Pressure” has been carefully revised. The revised manuscripts with or without correction track were loaded and for your easy check/editing purpose. A separated document answering every question raised by the reviewers one-by-one was shown as the following.
Reviewer 1: Comments and Suggestions for Authors
Tea is one of the most popular drinks worldwide and there is great interest in understanding effects of tea on blood pressure as well as on our general well-being. The review article by Li and colleagues attempted to summarize anti-hypertensive effects of different types of teas along with discussions of the pharmacological basis of such effects. Moderate English changes required
My comments are as follows:
Response: Thank you very much for positive comments of our manuscript! This revised manuscript has been checked by a native English speaker to improve English language.
1. Question: Overall structure of the review is poor. It does not talk anything about the methodology or analysis methods or controls. It is not clear why it was important to discuss “Main types of teas” under a separate headline, where the authors have talked a great deal about the classification of tea and their production methods. In addition, separating and discussing animal studies (headline 4, line 311) from those on tissues and cells (headline 5, line 442) has interrupted the natural flow of the discussion.
Answer:
Thank you very much for your deeply insight comments! Based on your suggestions, we added methodology or analysis methods or controls information in revised manuscript. Also, we provided comment for every selected human population study. In order to give these information together for easy reading, we provided the detailed information in Table 1 of revised manuscript. Please see these changes in the Table 1.
According to your suggestion, we deleted whole section 2 "Main types of tea". In this review manuscript, we will frequently mention tea types (green tea, black tea or oolong tea) and main tea chemical compounds for human and animal experiments. Therefore, we added several sentences in Introduction part. Please see detailed information on line 61-62 and line 64-67.
Based on your comment, in section 4, we focused on molecular mechanisms of tea regulating blood pressure, deleted “in tissues and cell cultures” in the sub title, and removed all mechanisms data from animal study (section 3) to section 4. Please see detailed information in our revised manuscript (line 546-909,section 4).
2. Question: Definition of hypertension is problematic. The authors did not specify what guidelines were followed for defining hypertension. The latest guidelines for hypertension published in November 2017 by ACC/AHA replaces previous guidelines. Mean BP is not consistent with the values for hypertension (e.g. heading 3.4). Use of term such as ‘mildly hypertensive’ is difficult to understand, as it does not have any numbers to specify this condition. If hypertension is more common in diabetic patients (as mentioned in 3.4) and tea does not have any effects on glucose tolerance, insulin secretion etc., how does it lower BP in this patients? Relevant discussion and commentary are missing throughout. Human intervention studies show only 1-3 mmHg reduction of SBP and DBP. However, data and relevant commentary is missing as to how such small reductions of BP lowers CV risk. There is a huge disparity between blood pressure lowering effects of tea in animal studies and human studies and, it is expected that the authors shed light on such observed differences.
Answer: Thanks for your kind suggestions! Based on the latest guidelines for hypertension published in November 2017 by ACC/AHA, we defined Stage 1 hypertension as a systolic blood pressure (SBP) of 130–139 mmHg or a diastolic blood pressure (DBP) of 80-89 mmHg, and Stage 2 hypertension as a systolic blood pressure (SBP) of ≥140 mmHg and/or a diastolic blood pressure (DBP) of ≥90 mmHg. Please see the detailed description in the text (Line 48-51).
Because mean BP is not consistent with the values for hypertension, we deleted all mean BP values in the revised manuscript. Also, based on your suggestion, ‘mildly hypertensive’ is difficult to understand, as it does not have any numbers to specify this condition. After careful comparation the values, we realized that‘mildly hypertensive’ is correlated to “Stage 1 hypertension”. Therefore, we changed all terms of ‘mildly hypertensive’ to “Stage 1 hypertension” in the revised manuscript.
According to previous research, hypertension is more common in diabetic patients. Hypertension is approximately twice as frequent in patients with diabetes compared with patients without the disease. [reference 95]. Recently, a systematic review and meta-analysis of randomized controlled trials involved in 27 studies (1898 participants) suggested that green tea may reduce fasting blood glucose (FBG) levels compared with placebo/water. Subgroup analysis showed that the effect of green tea on fasting blood glucose levels was significant only in studies with a mean age of < 55-years-old or Asian-based studies [96]. Previous researches reported that the potential beneficial effect of green tea on glucose metabolism may be mediated by epigallocatechin gallate (EGCG), the most abundant catechin present in green tea [97]. Waltner-Law et al. reported that EGCG reduces hepatic glucose production by increasing tyrosine phosphorylation of the insulin receptor and insulin receptor substrate-1 in H4IIE rat hepatoma cell [98]. Recent studies have also suggested that green tea and yellow tea increase insulin sensitivity and glucose metabolism, preventing progress of type 2 diabetes [99]. Ortsäter et al. also reported that EGCG preserves islet structure and enhances glucose tolerance in db/db mice [100]. Therefore, teas decrease diabetes-related hypertension might mediate by ameliorating diabetes complication, which prevents high glucose induced damages of vascular endothelial and smooth muscle cells and maintain normal dilation and contraction of vascular system. Based on your suggestion, we added this paragraph in the Discussion part of the text (line 1084-1100).
3. Question: Human intervention studies show only 1-3 mmHg reduction of SBP and DBP. However, data and relevant commentary is missing as to how such small reductions of BP lowers CV risk. There is a huge disparity between blood pressure lowering effects of tea in animal studies and human studies and, it is expected that the authors shed light on such observed differences.
Answer: Clinical data shows that a 5 mmHg blood pressure reduction can reduce the risk of stroke and ischemic heart disease by 34 and 21%, respectively [6,7]. Although no clinical data available to demonstrate CVD outcome of 1-3 mmHg reduction of SBP and DBP, such small reductions of BP may benefit Stage 1 hypertension patients to control BP at range. Based on animal models, teas treatment significantly decreased SBP and DBP. There is a huge disparity between blood pressure lowering effects of tea in animal studies and human intervention. This difference may due to simplicity of animal studies and complicity of human intervention. In animal experiment, genetic background (same strain), living environment (standard facility), dosage and duration of treatment can be strictly controlled. Therefore, it is easy to see BP lowing effect by teas. For human population studies, complicated social and genetic backgrounds, varying degrees of hypertensive, dietary intaking and physical activity all can affect BP lowing effect of teas. Therefore, tighter controls are needed to determine BP lowing effects in population studies. Based on your suggestion, we added this discuss in the Discussion part. Please see the last paragraph of Discussion part.
4. Question: The authors only described the anti-hypertensive effects of tea without any mention about other studies where tea has been shown to increase blood pressure. The review is one-sided with no consideration for the broader effects tea may have on human health and blood pressure.
Answer: Thank you very much for your insight comment! Actually, we have addressed this issue in the first paragraph of Discussion. Please see detailed information in the line 1041 to 1083.
Minor comments:
5. Question: The title of the article does not convey a clear message and therefore looks less attractive at first sight.
Answer: Thank you very much for your very good suggestion! After the careful consideration, we would like to change the title as " Effects and Mechanisms of Tea Regulating Blood Pressure: Evidences and Promises".
6. Question: Relevant reference is missing in many places. For example:
In page 1, line 36, ‘CVDs have overtaken cancer as the major cause of death……..”.
In page 1, line 38-39, ‘ Of these risk factors, high blood pressure is the most dangerous factor linked to CVDs death events.
There are so many places in the manuscript where appropriate references are missing. Also, references cited do not cover the major studies supporting that claim. For example:
In page 2, lines 46 – 50, there are more than just one reference, which should have been cited. These are just examples; the manuscript should be thoroughly revised for consistency.
Answer: Thanks you very much for your carefully checking! According to your suggestions, we added the relevant references at the places you mentioned. In addition, we carefully checked the whole text of this manuscript, and tried to add new references where needed. Please see the detailed changed in the revised manuscript!
7. Question: The authors presented a table for the human studies but it is missing some key information that the readers would be interested in. For example, it does not mention experiment design or conditions used for the studies or for analysis methods. There is no conclusion/comments either.
Answer: Thanks you very much for your very good suggestion! Based on your suggestion, we added key information such as “Methods of study”, selected participants and dosages to the Table 1. Also, based on experiment design, population size and analysis method, we added comment for every human intervention study. Please see detailed information in Table 1.
Reviewer 2 Report
Li et al. present, in the reviewed paper, an appraisal of current knowledge on the relationship between tea consumption and blood pressure regulation. Although the review is extensive and informative, there are several substantial issues that need to be addressed:
1. Chaotic citation assignment. Clearly, the citations were not processed in a very careful manner and in many cases, the reference numbers do not lead to the referred work. One example for many, on line 169 the authors refer to Tong et al [23] , in the Table 1 the same study has a reference no. [19] yet in the bibliography the real reference number is [25]!
2. A major shortcoming is that no method or criteria for including sources of information is presented. Were any papers of any quality used? What search terms and other constraints were used for acquisition, quality checking of papers to be considered?
3. There are many statements that need evidence or reference to back them up. For example, the authers define, on lines 43-44, hypertension as BP > 140/90 mmHg. Based on which guidelines/definitions? On the same page, most of the information on tea seems to be derived from a single review paper without proper referencing of hte original research to back the claims.
4. The Discussion and perspective section should be enhanced substantially. As the review text itself lacks the necessary layer provided by expertise of authors and in most parts represents rather an assembly of abstracts , this should be balanced in providing higher-level reflection of the gathered data from the clinical perspective (is a change of 1-2 mmHg in blood pressure truly clinically relevant?; issues with concentrations of active substances, pharmacogenetic aspects etc. etc.) as well as molecular one.
Author Response
Dear Editors and Reviewers:
Thank you very much for your letter and the referees’ reports! Based on your comments and suggestions, our manuscript entitled, “Effects and Mechanisms of Tea Regulating Blood Pressure” has been carefully revised. The revised manuscripts with or without correction track were loaded and for your easy check/editing purpose. A separated document answering every question raised by the reviewers one-by-one was shown as the following.
Reviewer 2: Comments and Suggestions for Authors
Li et al. present, in the reviewed paper, an appraisal of current knowledge on the relationship between tea consumption and blood pressure regulation. Although the review is extensive and informative, there are several substantial issues that need to be addressed. English language and style are fine/minor spell check required.
Response: Thank you very much for positive comments of our manuscript! This revised manuscript has been checked by a native English speaker to improve English language.
1. Chaotic citation assignment. Clearly, the citations were not processed in a very careful manner and in many cases, the reference numbers do not lead to the referred work. One example for many, on line 169 the authors refer to Tong et al [23] , in the Table 1 the same study has a reference no. [19] yet in the bibliography the real reference number is [25]!
Answer: Thanks you very much for your careful checking! We appreciated your time! We have corrected the wrong citations you mentioned. In addition, we made double checked the citation thought out manuscript, and make sure the citations are in the right place. Please see the detaled information in the revised manuscript.
2. A major shortcoming is that no method or criteria for including sources of information is presented. Were any papers of any quality used? What search terms and other constraints were used for acquisition, quality checking of papers to be considered?
Answer: Thank you very much for your insight comment! Like writing very review paper, we first tried to collect all the published papers available on this area. Then, we judged the papers quality based on the experiment design, population size and analysis method, as well as good credit for the original journals. Based on your suggestion and easy for the readers, we added these key information such as “Methods of study”, selected participants and dosages used to the Table 1. Also, we added comment for quality of every human intervention study. Please see detailed information in Table 1.
3. There are many statements that need evidence or reference to back them up. For example, the authers define, on lines 43-44, hypertension as BP > 140/90 mmHg. Based on which guidelines/definitions? On the same page, most of the information on tea seems to be derived from a single review paper without proper referencing of the original research to back the claims.
Answer: Thank you very much for your very good comments! Based on the latest guidelines for hypertension published in November 2017 by ACC/AHA, we defined Stage 1 hypertension as a systolic blood pressure (SBP) of 130–139 mmHg or a diastolic blood pressure (DBP) of 80-89 mmHg, and Stage 2 hypertension as a systolic blood pressure (SBP) of ≥140 mmHg and/or a diastolic blood pressure (DBP) of ≥90 mmHg. We cited this paper as reference 8. Please see the detailed description in the text (Line 48-51).
According to your comment, we realized that there are not appropriate for this manuscript to write the classification of tea and their produce methods in detail. We deleted whole section 2 "Main types of tea". In this review manuscript, afterwards we will frequently mention tea types (green tea, black tea or oolong tea) and main tea chemical compounds for human and animal experiments. Therefore, we added several sentences in Introduction part. Please see detailed information on line 57-58 and line 60-63.
4. The Discussion and perspective section should be enhanced substantially. As the review text itself lacks the necessary layer provided by expertise of authors and in most parts represents rather an assembly of abstracts , this should be balanced in providing higher-level reflection of the gathered data from the clinical perspective (is a change of 1-2 mmHg in blood pressure truly clinically relevant?; issues with concentrations of active substances, pharmacogenetic aspects etc. etc.) as well as molecular one.
Answer: Thank you very much for your invaluable comments! As your suggestion, we enhanced discussion part by adding additional two paragraph to address tea preventing diabetes associated hypertension and clinically relevant of small reduction (1-3 mmHg) in blood pressure. Clinical data shows that a 5 mmHg blood pressure reduction can reduce the risk of stroke and ischemic heart disease by 34 and 21%, respectively [6,7]. Although no clinical data available to demonstrate CVD outcome of 1-3 mmHg reduction of SBP and DBP, such small reductions of BP may benefit Stage 1 hypertension patients to control BP at normal range. Please see detailed information at the last paragraph of Discussion part.
Round 2
Reviewer 1 Report
While the manuscript still has scope for improvement, most of the concerns have been reasonably addressed.